# Bayesian Active Learning with Fully Bayesian Gaussian Processes

## Abstract

The bias-variance trade-off is a well-known problem in machine learning that only gets more pronounced the less available data there is. When data is scarce, such as in metamodeling, active learning, and Bayesian optimization, neglecting this trade-off can cause inefficient and non-optimal querying, leading to unnecessary data labeling. In this paper, we focus on metamodeling with active learning and the canonical Gaussian Process (GP). We recognize that, for the GP, the bias-variance trade-off regulation is made by optimization of the two hyperparameters: the length scale and noise-term. Considering that the optimal mode of the joint posterior of the hyperparameters is equivalent to the optimal bias-variance trade-off, we approximate this joint posterior and utilize it to design two new acquisition functions. The first one is a mode-seeking Bayesian variant of Query-by-Committee (B-QBC), and the second is simultaneously mode-seeking and minimizing the predictive variance through a Query by Mixture Gaussian Processes (QB-MGP) formulation. Across seven simulators, we empirically show that B-QBC outperforms the benchmark functions, whereas QB-MGP is the most robust acquisition function and achieves the best accuracy with the fewest iterations. We generally show that incorporating the bias-variance trade-off in the acquisition functions mitigates unnecessary and expensive data labeling.

## 1 Introduction

Gaussian Processes (GPs) are the canonical models to use for Bayesian optimization and metamodeling (Snoek et al., 2012; Gramacy, 2020). GPs are well-known for their ability to deal with small to medium size data sets as well as balancing complexity and regularization - together with their inherent ability to handling uncertainties - this makes them ideal in such applications (Williams & Rasmussen, 2006). In both cases, often only limited data is accessible so the natural balance between complexity and regularization helps prevent severe overfitting, additionally making the model flexible enough to model nonlinear functions. Likewise, the quantification of uncertainty is commonly used in the acquisition function to guide the Bayesian optimization algorithms and the active learning schemes that are almost inevitable to efficiently build a metamodel.

On the other hand, it is not a flawless procedure to use GPs to guide the two schemes. The same GP is used as both *predictor* and *guide*, and thus the choice of predictor will affect the guide, and vice versa. Trivially, the more data there is available for modeling, the less pronounced this problem will be. However, in the context of both Bayesian optimization and active learning, where the data sets tend to be rather small, a wrong predictor can result in misguidance, thus hindering the performance and efficiency. In this paper, we mitigate this problem by not only focusing on a single predictive model but consider many predictive models through multiple model hypotheses at once.

A GP is typically fitted through evaluation of the marginal likelihood, which automatically incorporates a trade-off between complexity and regularization (Williams & Rasmussen, 2006). However, when the data is scarce, it is more challenging to choose the appropriate trade-off, and different configurations of the hyperparameters of the GP can give rise to distinct fits. This is illustrated in Figure 1, where two seemingly reasonable fits might guide the schemes quite differently. The problem is directly related to the well-known bias-variance trade-off, although we reformulate it as a balance between modeling the data as signal or noise. For the common stationary covariance functions of GPs, e.g. the radial-basis function or the Matérn class of functions, together with a Gaussian

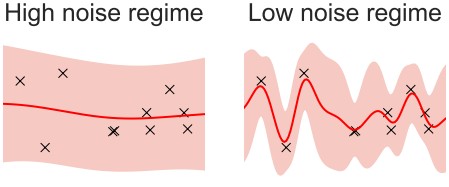

Figure 1: Two GPs with different hyperparameters. The left plot shows a GP with high noise and a long length scale, and the right plot shows a GP with low noise and a short length scale.

Figure 2: Joint posterior of the two hyperparameters length scale and noise. The posterior is multimodal with a low noise, high signal and a high noise, low signal mode.

likelihood, this is directly reflected in the two hyperparameters *length scale* $\ell$ and *noise* $\sigma_\varepsilon^2$. If the data is modeled primarily as noise, both $\sigma_\varepsilon^2$ and $\ell$ are large, whereas if the data is modeled primarily as a signal, both $\sigma_\varepsilon^2$ and $\ell$ are small. In the case of limited data, the joint posterior distribution of the two hyperparameters is likely to be characterized by two modes, as illustrated in Figure 2 for the data in Figure 1. In that case, it is difficult to tell whether it is best to model the data as noise or signal, and a wrong choice of mode will imply non-optimal guidance from the acquisition function.

The literature suggests handling this problem with clever initializations of the hyperparameters Williams & Rasmussen (1996) or by favoring small $\ell$ and $\sigma_\varepsilon^2$ by either always initializing hyperparameters in the low noise regime or by applying strong priors (Gramacy, 2020). However, none of these approaches directly address the core problem: which mode to choose? Ideally, this should be answered with prior information about the problem, although typically that is not available, making these approaches less practical.

In this paper, we follow a general approach and assume no prior knowledge about the kernel or hyperparameters. We consider multiple model hypotheses by replacing the fitting procedure of the marginal likelihood with Markov Chain Monte Carlo (MCMC) sampling. Lalchand & Rasmussen (2020) show that with fixed, medium-sized data sets and carefully chosen kernels, it is beneficial to fit GPs with MCMC instead of by maximizing the marginal likelihood. We show that the same is true for very small data sets and with a general kernel. Our main contribution is the proposal of two new acquisition functions for active learning that utilizes the extra information from the hyperparamters' posteriors estimated by MCMC to seek the most reasonable mode alongside minimizing the predictive variance. We show that the two acquisition functions are more accurate and robust than other common functions across multiple benchmark simulators used in the related literature.

## 2 RELATED WORK

The two proposed acquisition functions are specifically designed for active learning schemes for regression tasks. In this section, we lay out the related work, although the specific acquisition functions used for comparison purposes in the experiments are described in detail in section 4. Further, in this section, we cover the essentials of Query-by-Committee and Gaussian Mixture Models, as these constitute the backbone of the proposed acquisition functions.

**Active Learning** The main idea of active learning (AL) is to *actively* choose a new data point to label and add to the current training data set (Settles, 2009). In the context of metamodeling, new data is often added sequentially, i.e., one data point at a time (Gramacy, 2020), but in other applications, it can be beneficial to query batches of data instead (Kirsch et al., 2019).

The acquisition functions can be divided into model-based and model-free functions, where the former utilize information from the model and the latter do not (O'Neill et al., 2017). Both types of functions seek to minimize the expected predictive loss of the model. Another approach is to minimize the number of possible models. Houlsby et al. (2011) divide the active learning acquisition functions into being based on either decision or information theory. Decision-based functions seek to minimize the expected predictive loss of the model in the hope of maximizing the performance

on the test set. Information-theoretic-based functions instead try to reduce the number of possible models, for example, through the KL-divergence or Shannon entropy (Houlsby et al., 2011).

It is not straightforward to use information-theoretic acquisition functions. However, if you have access to the posterior of the model's parameters, Houlsby et al. (2011) have derived the algorithm *Bayesian Active Learning by Disagreement* (BALD), which can be applied in general. Generally, BALD seeks the data point that maximizes the decrease in the expected posterior entropy of the parameters.

**Query-by-Committee**   The Query-by-Committee (QBC) is a specific acquisition function that was originally proposed for classification tasks (Seung et al., 1992). It aims to maximize the disagreement among the committee to get the highest information gain and minimize the version space, which is the set of model hypotheses aligned with the training data. The construction of the committee is the core component of QBC since it is the committee's ability to accurately and diversely represent the version space that gives rise to informative disagreement criteria (Settles, 2009).

QBC can also be applied for regression problems. Krogh & Vedelsby (1995) construct the members of the committee by random initializations of the weights in the neural networks. RayChaudhuri & Hamey (1995) apply bagging and train the members on different subsets of the data set. In general, QBC constructed by bagging has been used as a benchmark with mixed results (Cai et al., 2013; Wu, 2018; Wu et al., 2019). Burbidge et al. (2007) show that the less noise there is in the output, the better QBC is compared to random querying. They also highlight the fact that with a misspecified model, QBC might perform worse than random querying. None of these approaches explore the usage of MCMC samples of the posterior to construct a committee. To the best of our knowledge, we are the first to propose QBC based on model hypotheses drawn from the posterior.

**Gaussian Process as a Gaussian Mixture Model**   Mixture models have recently been applied in active learning for classification tasks. Iswanto (2021) propose to use Gaussian Mixture Models (GMMs) with active learning, where they design a specific acquisition function that queries the data point that maximizes the expected likelihood of the model. Zhao et al. (2020) use a mixture of GPs in active learning, where each component is fitted to a subset of the training set.

The combination of GMMs and GPs have previously been explored for static data sets. Chen & Ren (2009) investigate regression tasks and apply bagging, where they repeatedly randomly sample data points from the training set to construct new subsets to get GPs fitted to different data. Among other combination rules, they combine the predictive posteriors of these GPs into GMMs. They obtain even better performance by weighting the models by the predictive uncertainty such that models with high uncertainty are given smaller weights, and vice versa. Instead of using bagging, we construct the multiple GPs by using the MCMC samples of the hyperparameters' joint posterior, and then obtain a natural weighting of the GPs: the GMM will consist of more GPs with hyperparameters close to the modes than hyperparameters far away. The formal procedure is given in section 4. To the best of our knowledge, we are the first to combine GMMs and GPs in this manner.

## 3   GAUSSIAN PROCESSES

The Gaussian Processes (GPs) are the central models in this work. In this section, we give a brief overview of GPs before covering the Fully Bayesian GPs. For a thorough description of GPs, we refer to Williams & Rasmussen (2006).

**Gaussian Processes**   A Gaussian Process (GP) is a stochastic function fully defined by a mean function $m(\cdot)$ and a covariance function (often called a kernel) $k(\cdot, \cdot)$. Given the data $(X, y) = \{x_i, y_i\}_{i=1}^N$, where $y_i$ is the corrupted observations of some latent function values $f$ with Gaussian noise $\varepsilon$, i.e., $y_i = f_i + \varepsilon_i$, $\varepsilon_i \in \mathcal{N}(0, \sigma_\varepsilon^2)$, a GP is typically denoted as $\mathcal{GP}(m_f(x), k_f(x, x'))$. It is common practice to set the mean function equal to the zero-value vector and thus, the GP is fully determined by the kernel $k_f(x, x')$. For short, we will denote the kernel $K_\theta$, which explicitly states that the kernel is parameterized with some hyperparameters $\theta$. Given the optimal hyperparameters, the predictive posterior for unknown test inputs $x^*$ is given by $p(f^* | \hat{\theta}, y, X, X^*) = \mathcal{N}(\mu^*, \Sigma^*)$ with

$$\mu^* = K_\theta^* \left( K_\theta + \sigma_\varepsilon^2 \mathbb{I} \right)^{-1} y \quad \text{and} \quad \Sigma^* = K_\theta^{\star\star} - K_\theta^* \left( K_\theta + \sigma_\varepsilon^2 \mathbb{I} \right)^{-1} K_\theta^{\star\top} \tag{1}$$

where $K_\theta^{\star\star}$ denotes the covariance matrix between the test inputs, and $K_\theta^*$ denotes the covariance matrix between the test inputs and training inputs.

**Covariance matrix** We use the canonical kernel automatic relevance determination (ARD) Radial-basis function (RBF) given by $k(x, x') = \exp\left(-||x - x'||^2/2\ell^2\right)$ where $\ell$ is a vector of length scales $\ell_1, ..., \ell_d$, one for each input dimension. Often the kernel is scaled by an output variance but here we fix it to one and solely focus on the two other hyperparameters: length scale and noise-term. The noise-term $\sigma_\varepsilon^2$ is integrated into the kernel with an indicator variable by adding the term $\sigma_\varepsilon^2 \mathbb{I}_{\{x=x'\}}$ to the current kernel.

**Fully Bayesian Gaussian Processes (FBGP)** A FBGP extends a GP by putting a prior over the hyperparameters $\theta \sim p(\theta)$ and approximate their full posteriors. The joint posterior is then given by

$$p(f, \theta|y, X) \propto p(y|f)p(f|\theta, X)p(\theta) \tag{2}$$

and the predictive posterior for the test inputs $X^*$ is

$$p(f^*|y) = \iint p\left(f^*|f, \theta\right) p(f|\theta, y)p(\theta|y)df\,d\theta \tag{3}$$

where the conditioning on $X$ and $X^*$ have been omitted for brevity. The inner integral reduces to the predictive posterior given by a normal GP. However, the outer integral remains intractable and is approximated with MCMC inference as

$$p\left(f^*|y\right) = \int p\left(f^*|y, \theta\right) p(\theta|y)d\theta \quad \simeq \quad \frac{1}{M}\sum_{j=1}^{M} p\left(f^*|y, \theta_j\right), \quad \theta_j \sim p(\theta|y) \tag{4}$$

As well known in machine learning modeling, there is no free lunch. Adapting the hyperparameters of a FBGP is computationally expensive compared to the approach with GPs and maximum likelihood estimates. However in Bayesian optimization and active learning, the computational burden for querying a new data point will often be of magnitudes higher.

**Evaluation and Fitting of GPs** A GP is typically fitted by maximizing the marginal likelihood with gradient descent, where the marginal likelihood can be computed analytically. For the FBGP, the inner integral is intractable and is therefore optimized with approximate inference using the self-tuning method of Monte Carlo Markov Chain (MCMC) called the No-U-Turn-Sampler (NUTS) (Hoffman & Gelman, 2014). The MCMC inference makes the full posterior of the hyperparameters available, making it possible to extend the point estimates to a joint distribution. In the experiment, we apply acquisition functions that use the full posterior distribution; however, since the experiments are designed to benchmark the acquisition functions, we always use the mode of the joint posterior for prediction and evaluation.

## 4 ACTIVE LEARNING

In this section, we lay out the most common acquisition functions and then propose first a Bayesian variant of Query-by-Committee and secondly an extension motivated by Gaussian Mixture Models, which seek to minimize both the predictive variance and the number of model hypotheses. All the acquisition functions choose a data point $x$ among the possible data points in the unlabeled pool $U$.

### 4.1 CLASSIC ACTIVE LEARNING

Many classic active learning acquisition functions are based on the model's uncertainty and entropy (Settles, 2009; Gramacy, 2020). We use the most common acquisition function based on the predictive entropy and denoted *Active Learning MacKay* (ALM) (MacKay, 1992).

**Entropy (ALM)** For a Gaussian distribution, the Shannon entropy $H[\cdot]$ is proportional to the predictive variance $\sigma^2(x)$, i.e. $H[x] = \frac{1}{2}\ln((2\pi\sigma^2(x))$, so a new data point $x_{new}$ is queried as

$$x_{new} = \arg\max_x \sigma^2(x) \tag{5}$$

Intuitively, ALM queries the data point, where the uncertainty of the prediction is the highest.

### 4.2 BAYESIAN ACTIVE LEARNING

If we have access to the posterior of the model's parameters, we can utilize acquisition functions with an extra Bayesian level, giving rise to Bayesian active learning. The posterior of the hyperparameters of an FBGP can be estimated with MCMC such that GPs with different pairs of the kernel parameters, length scale, and noise can be drawn, i.e., $\ell, \sigma_\varepsilon^2 \sim p(\theta|y, X)$. The following four acquisition functions all utilize this information and are approximated using the samples from the MCMC, where we compute $p(f^*|\hat{\theta}, y, X, X^*)$ using equation 4.

**Entropy (B-ALM)** In the Bayesian setting, we can use the information from the full posteriors instead of only using the point estimates. The Bayesian variant of ALM (B-ALM) is then given as

$$x_{new} = \arg\max_x \mathbb{E}_{\theta \sim p(\theta|D)}[\sigma_\theta^2(x)|\theta] \tag{6}$$

**Bayesian Active learning by Disagreement (BALD)** Houlsby et al. (2011) propose to find the data point that maximizes the decreases in the expected posterior entropy of the hyperparameters. They rewrite the objective from computing entropies in the parameter space to the output space by observing that it is equivalent to maximizing the conditional mutual information between the parameters and unknown output $I[\boldsymbol{\theta}, y|x, \mathcal{D}]$, where $(x, y)$ is the new data point and $\mathcal{D}$ is the training data. The acquisition function is denoted Bayesian Active Learning by Disagreement (BALD) and is given by:

$$x_{new} = \arg\max_{\boldsymbol{x}} I[\boldsymbol{\theta}, y|x, \mathcal{D}] = \arg\max_{\boldsymbol{x}} H[y|\boldsymbol{x}, \mathcal{D}] - \mathbb{E}_{\boldsymbol{\theta} \sim p(\boldsymbol{\theta}|\mathcal{D})}[H[y|\boldsymbol{x}, \boldsymbol{\theta}]] \tag{7}$$

BALD was originally derived for non-parametric discriminative models but has recently been extended to batches and deep learning with great success (Kirsch et al., 2019). Despite the success, we show that BALD is not the optimal way to query new data points when considering GPs for regression tasks.

**Bayesian Query-by-Committee (B-QBC)** Motivated by reducing the number of possible models and with the multimodal posterior of the parameters in mind, we propose a new acquisition function that seeks the correct mode. More precisely, we propose to use the joint posterior of the parameters obtained through MCMC to draw multiple models and then query a new data point where the mean predictions $\mu_\theta(x)$ of these models disagree the most. Since the models are drawn from the parameters' posterior, the collection of models is dominated by models near the posterior modes. Querying the data point that maximizes the disagreement between the models' predictions corresponds to querying the data point providing the highest information about the true mode. Thus, this can be seen as a mode-seeking Bayesian Query-by-Committee (B-QBC). Given that $\overline{\mu}(x)$ is the average mean function, B-QBC is given as

$$x_{new} = \arg\max_x \mathbb{E}_{\theta \sim p(\theta|D)}[(\mu_\theta(x) - \overline{\mu}(x))^2|\theta] \tag{8}$$

**Query by Mixture of Gaussian Processes (QB-MGP)** We extent the mode-seeking B-QBC to consider the predictive variance as well. We denote the new acquisition function *Query by Mixture of Gaussian Processes* (QB-MGP) since it can be motivated by the fact that each prediction of the FBGP can be written as a Gaussian Mixture Model (GMM). More formally, we propose to use the MCMC samples to make each prediction into a GMM, yielding the predictive posterior given as

$$p(f^*|y) = \int p(f^*|y, \theta) p(\theta|y) d\theta \quad \simeq \quad \frac{1}{M} \sum_{j=1}^{M} p(f^*|y, \theta_j), \quad \theta_j \sim p(\theta|y) \tag{9}$$

This hierarchical predictive posterior is a mixture of $M$ Gaussians with mean $\mu_{GMM}$ and covariance matrix $K_{GMM}$ defined as (Lalchand & Rasmussen, 2020):

$$\mu_{GMM} = \frac{1}{M} \sum_{i=1}^{M} \mu_{\theta_i} \quad \text{and} \quad K_{GMM} = \frac{1}{M} \sum_{i=1}^{M} K_{\theta_i} + \frac{1}{M} \sum_{i=1}^{M} (\mu_{\theta_i} - \mu_{GMM})^2 \tag{10}$$

Finding the data point that maximizes the variance of the GMM is now equivalent to simultaneously considering the B-ALM and B-QBC:

$$x_{new} = \arg\max_x \mathbb{E}_{\theta \sim p(\theta|D)}[\sigma_\theta^2(x)|\theta] + \mathbb{E}_{\theta \sim p(\theta|D)}[(\mu_\theta(x) - \overline{\mu}(x))^2|\theta] \tag{11}$$

Table 1: Stochastic simulators used in the experiments.

| | Simulator | $d$ | Noise $\sigma_\varepsilon$ | Input space | Previously used in |
|---|---|---|---|---|---|
| 1 | Sine: $\sin(6x)$ | 1 | 0.2 | $[0, 5]$ | |
| 2 | Gramacy1d | 1 | 0.1 | $[0.5, 2.5]$ | Gramacy & Lee (2012) |
| 3 | Higdon1d | 1 | 0.1 | $[0, 20]$ | Gramacy & Lee (2009); Gramacy (2020) |
| 4 | Motorcycle | 1 | Het* | $[0, 60]$ | Silverman (1985); Gramacy & Lee (2008); Gramacy (2020) |
| 5 | Gramacy2d | 2 | 0.05 | $[-2, 6]^2$ | Gramacy & Lee (2009; 2012); Sauer et al. (2020) |
| 6 | Branin | 2 | 11.32 | $[-5, 10] \times [0, 15]$ | Keane et al. (2008); Picheny et al. (2013); Cole et al. (2021) |
| 7 | Ishigami | 3 | 0.187 | $[-\pi, \pi]^3$ | Marrel et al. (2009); Cole et al. (2021) |
| 8 | Hartmann | 6 | 0.0192 | $[0, 1]^6$ | Picheny et al. (2013); Cole et al. (2021) |

*Heteroscedastic noise

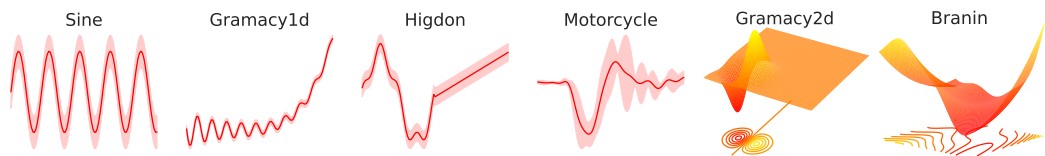

Figure 3: Visualization of the simulators (excl. Ishigami and Hartmann due to the dimensionality).

## 5 EXPERIMENTS

In this section, we start by introducing a simple simulator denoted *Sine* (see Figure 3), which we use to show that with few data points and no prior knowledge, it can be catastrophic to use a GP fitted by MAP since it does not find the optimal mode. We show that the FBGP partly solves the problem by utilizing the information from the full posterior. Additionally, we show that both B-QCB and QB-MGP outperform the other acquisition functions. Both results are shown in Figure 4.

We benchmark the performance of the FBGP and the two proposed acquisition functions with the standard fitting procedures of maximum a posteriori (MAP) and the standard acquisition functions based on the entropy, i.e., ALM, B-ALM, and BALD, on various classic simulators used in recent literature on GPs and active learning. They are all listed in Table 1, and those with less than three inputs are shown in Figure 3[1]. The motorcycle simulator is created by fitting a variational GP (Hensman et al., 2015) to the motorcycle accident data set (Silverman, 1985). The evaluation metrics are the negative log marginal likelihood (NLML) and root mean square error (RMSE). The performance for each the active learning iteration is shown, but for a comparable overview, we focus on a reference iteration that follows a convergence criterion: when the best acquisition function (based on NLML) has reached convergence, we label that iteration as reference (for that specific simulator). If no convergence is reached, we use iteration number 100 as reference. For these evaluations, the introduced toy simulator is excluded, and we only consider simulators used in the literature previously.

In the experiments, we use a zero-mean GP with an ARD RBF kernel. In each iteration of the active learning loop, the inputs are rescaled to the unit cube $[0, 1]^d$, and the outputs are standardized to have zero mean and unit variance. Following Lalchand & Rasmussen (2020), we give all the hyperparameters relatively uninformative $\mathcal{N}(0, 3)$ priors in log space. The initial data sets consist of three data points chosen by maximin Latin Hypercube Sampling, and in each iteration, one data point is queried. The unlabeled pool $U$ consists of the input space discretized into 100 equidistant points along each dimension. If $U$ contains more than $10,000$ data points, we randomly sample a subset of $10,000$ data points in each iteration and use that as the new pool. For the MAP estimation, the optimal hyperparameters are estimated by gradient descent using Adam (Kingma & Ba, 2015) with a learning rate of 0.1. The inference in FBGP is carried out using NUTS (Hoffman & Gelman, 2014) in Pyro (Bingham et al., 2018) with five chains and 500 samples, including a warm-up period with 200 samples. The remaining 1500 samples are all used for the Bayesian acquisition functions. All experiments are repeated ten times with different initial data sets. The models are implemented in GPyTorch (Gardner et al., 2018), and all code can be found at `www.available/later.com`.

---

[1]All of them except number 1 and 4 can be found at `https://www.sfu.ca/~ssurjano/`.

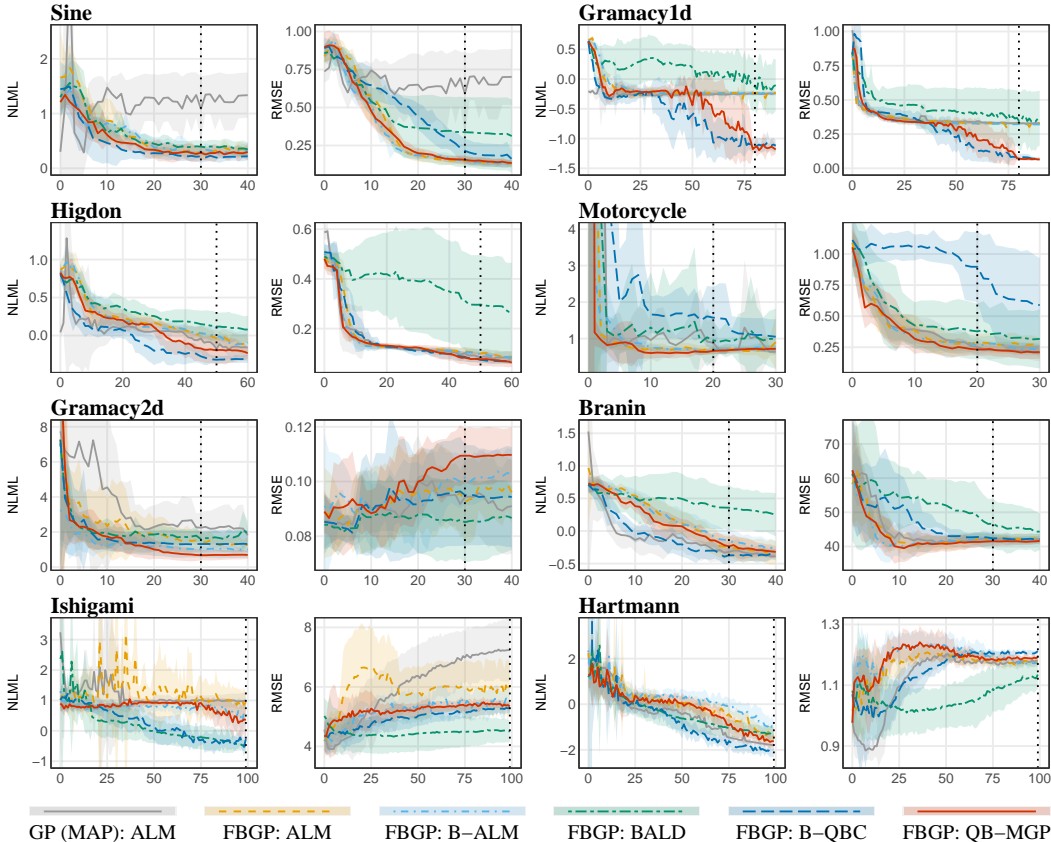

Figure 4: Performance across the 10 runs $\pm 1$ standard deviation. The $x$-axis represents the number of iterations, and the vertical dotted lines show the reference iterations (also listed in Table 2).

## 5.1 EXPERIMENTS WITH 8 SIMULATORS

We benchmark the acquisition functions on the *Sine* simulator and seven classic simulators. We divide the simulators into subgroups and describe how the functions works in different noise settings, complexities, and with multiple inputs, effectively encompassing five distinct modeling scenarios.

**Simple simulator** The simulator *Sine* is constructed to show the advantage of FBGPs compared to GPs. It also shows the benefit of applying B-QBC and QB-MGP compared to the other acquisition functions. Considering the NLML, B-QBC is slightly better than the other functions, although when we look at the RMSE, QB-MGP is the overall best choice. This shows the gain of reducing the predictive variance alongside the number of model hypotheses.

**Noise or signal** This simulator *Gramacy1d* has previously been used to study the effect of the noise-term in GPs (Gramacy & Lee, 2012). The simulator has a periodic signal that is hard to reveal if the data points are not queried cleverly. Both B-QBC and QB-MGP reach convergence simultaneously, while the other acquisition functions struggle in distinguishing noise from the signal.

**Linear and non-linear output regions** The two simulators *Higdon* and *Gramacy2d* have been used to illustrate cases where GPs struggle in modeling the data due to the output signal having both linear and non-linear regions (Gramacy & Lee, 2009). Our experiments on these simulators show the performance of the acquisition functions when the GP is an inadequate choice of model. Querying data points in the linear and non-linear regions will yield a GP with a longer and shorter length scale, respectively. On Higdon, it is seen that both B-QBC and QB-MGP balance the sampling since the corresponding NLML and RMSE are low. On Gramacy2d, QB-MGP is achieving the highest RMSE due to favoring to capture the small region with a non-linear signal compared to the vast linear region. Overall, these results show that when the GP is inadequate to model the data, both B-QBC and QB-MGP perform better.

Table 2: The likelihood-ratios and the relative change in RMSE (%) compared to the best performing function averaged over 10 runs.

| | Gramacy1d | Higdon | Motorcycle | Gramacy2d | Branin | Ishigami | Hartmann | Overall performance | | |
|---|---|---|---|---|---|---|---|---|---|---|
| Dimensions | 1 | 1 | 1 | 2 | 2 | 3 | 6 | | | |
| Iteration | 80 | 50 | 20 | 30 | 30 | 100 | 100 | Mean | Median | Min |
| Likelihood-ratios | | | | | | | | | | |
| GP (MAP): ALM | -1.79 | -0.29 | -0.44 | -3.14 | -0.11 | -2.96 | -0.64 | -1.34 | -0.64 | -3.14 |
| FBGP: ALM | -1.82 | -0.57 | -0.04 | -1.62 | -0.28 | -2.88 | -1.36 | -1.22 | -1.36 | -2.88 |
| FBGP: B-ALM | -1.80 | -0.60 | -0.07 | -0.69 | -0.45 | -2.20 | -2.01 | -1.12 | -0.69 | -2.20 |
| FBGP: BALD | -2.04 | -0.87 | -0.58 | -1.84 | -1.46 | **0** | -1.66 | -1.21 | -1.46 | -2.04 |
| FBGP: B-QBC (Ours) | -0.04 | **0** | -1.82 | -1.28 | **0** | -0.45 | **0** | -0.51 | **-0.04** | -1.82 |
| FBGP: QB-MGP (Ours) | **0** | -0.25 | **0** | **0** | -0.28 | -1.71 | -0.99 | **-0.46** | -0.25 | **-1.71** |
| Relative change in RMSE (%) | | | | | | | | Mean | Median | Max |
| GP (MAP): ALM | 366 | 8 | 9 | 11 | 1 | 61 | 4 | 65.7 | 9 | 366 |
| FBGP: ALM | 359 | 35 | 25 | 9 | 2 | 34 | 4 | 66.9 | 25 | 359 |
| FBGP: B-ALM | 352 | 32 | 14 | 19 | 1 | 22 | 3 | 63.3 | 19 | 352 |
| FBGP: BALD | 387 | 299 | 63 | **0** | 12 | **0** | **0** | 108.7 | 12 | 387 |
| FBGP: B-QBC (Ours) | 15 | **0** | 286 | 12 | 2 | 17 | 7 | 48.4 | 12 | 286 |
| FBGP: QB-MGP (Ours) | **0** | 3 | **0** | 29 | **0** | 19 | 5 | **8.0** | **4** | **29** |

**Heteroscedastic noise** The experiments on the *Motorcycle* simulator explore how the active learning acquisition functions perform when the simulator has heteroscedastic noise, but we model it with a homoscedastic GP. Most conspicuous is the poor performance of B-QBC, indicating that this is misled by the heteroscedastic noise. However, the combination of B-ALM and B-QBC in QB-MGP gives the best performance. This agrees with our expectation that QB-MGP seems to be more robust than B-QBC to different types of noise.

**Multiple inputs** To evaluate the performance on higher dimensions, we consider the smooth 2d *Branin* simulator, the strongly non-linear 3d *Ishigami* simulator, and the 6d *Hartmann* simulator with six local minima. BALD underperforms on Branin, but the other acquisition functions have similar performance. Looking at all the active learning iterations, the GP fitted with MAP seems to give the best trade-off between the NLML and RMSE, indicating that the regular GP with ALM is the best for smooth problems. For Ishigami and Hartmann, 100 iterations are not enough for convergence, although B-QBC achieves the lowest NLML, whereas BALD reaches the lowest RMSE.

## 5.2 THE OVERALL PERFORMANCE

We now provide a performance overview by evaluating the acquisition functions at the iteration in which the best one has converged in terms of NLML. We both evaluate the NLML and RMSE, but to clearly see the difference in performance, we compute the likelihood-ratios for the NLML and the relative change in RMSE to the best performing acquisition function. In that way, we can summarize the performance across the different simulators. The likelihood-ratios and relative changes in RMSE are listed in Table 2, and the actual values are found in Appendix, Table 3.

Considering the likelihood-ratios, B-QBC and QB-MGP are the two best acquisition functions for six out of the seven simulators. The median performance of B-QBC is slightly better than QB-MGP, but both the mean and the minimum value of QB-MGP is better, indicating that QB-MGP is more robust across the simulators. Regarding the relative changes in RMSE, BALD and QB-MGP are the best performing functions. Overall, QB-MGP achieves both the lowest mean, median, and maximum value, clearly stating that QB-MGP is the most accurate and robust acquisition function across the simulators. For a visual representation, the box plots of the results are shown in Figure 5. B-QBC and QB-MGP are the two best acquisition functions based on the NLML. Regarding the RMSE, the B-QCB is lacking behind QB-MGP, and we see that QB-MGP benefits for taking the predictive uncertainty into account.

## 5.3 SCOPE & LIMITATIONS

In applications with active learning, it is not only the performance at convergence that is of interest since we often have a limited computational budget. However, in general, we observe that the relative performances between the acquisition functions do not vary significantly doing the ac-

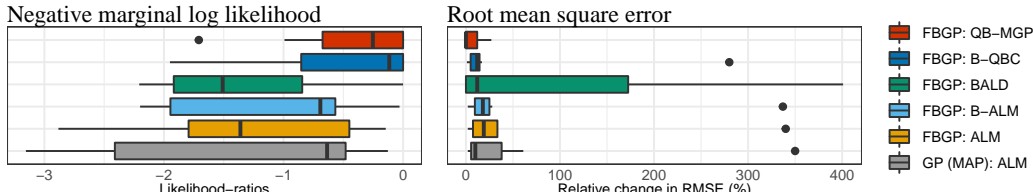

Figure 5: Visualization of likelihood-ratios and relative change in RMSE (%).

tive learning iterations, and thus we believe that the evaluations at the reference points provide an accurate evaluation.

The Bayesian acquisition functions utilize the full estimated joint posterior to choose a new data point. Similarly, we could also use the posterior for prediction instead of only using the mode. Then the predictive posterior is given as for the mixtures of Gaussian in equation 10. However, since we focus on benchmarking the acquisition functions and not the models, we believe that using the mode is more accurate when comparing it to the regular GP since we only change the fitting procedure.

This paper is based on empirical results that are dependent on the specific simulators. It would be infeasible to apply these acquisition functions to all previously used simulators, but we hope that our choice of diverse and distinct classic simulators is representative of the engineering problems occurring in the real world. Nonetheless, we encourage the reader to test other simulators by using our code at `www.available/later.com`.

## 6    CONCLUSION

In this paper, we propose two active learning acquisition functions: Bayesian Query-by-Committee (B-QBC) and Query by a Mixture of Gaussian Processes (QB-MGP), both of which are suited for fully Bayesian GPs. They are designed to explicitly handle the well-known bias-variance trade-off by optimization of the GP's two hyperparameters, length scale and noise-term. We empirically show that they query new data points more efficiently than previously used acquisition functions. Across seven classic simulators, which cover different types of noise, complexity and number of inputs, we show that QB-MGP is the most robust function and achieves the best accuracy with fewest iterations.

The incorporation of domain knowledge and expert guidance regarding the simulator under study can be a decisive factor in a successful metamodeling strategy. However, in many practical situations, such a priori domain expertise may not be readily accessible or even translatable into the functional structure of the metamodel as useful modeling information. On these occasions, generic tools that are robust enough to handle a plethora of diverse simulation output behaviors are prudently advisable. On the one hand, we aim at maintaining an economical approach to the problem of exploring the simulation input space. On the other hand, the final fitting performance of the metamodel must be equally taken into utmost account. Simply discarding one of these two aspects might well end up rendering the active learning-based metamodeling strategy not only computationally unsustainable but also effectively counterproductive. To this end, we believe that the proposed acquisition functions properly address both concerns, while being entirely independent of any prior understanding of the underlying output distributions of the simulator.

In terms of future work, we plan to develop new acquisition functions, which constitute, in the short-term, variations of those explored herein. Then, we aim to apply the proposed approach in the context of larger and realistic simulation models addressing real-world problems. Eventually, our ultimate goal is to provide practitioners with an auxiliary tool to map the simulators' output behaviors in a more efficient manner.

### ACKNOWLEDGMENTS

This work was supported by NOSTROMO, framed in the scope of the SESAR 2020 Exploratory Research topic SESAR-ER4-26-2019, funded by SESAR Joint Undertakingthrough the European Union's Horizon 2020 research and innovation programme under grant agreement No 892517.

REPRODUCIBILITY STATEMENT

In this work, we strove to make the results as reproducible as possible. To this end, besides using open and free source software (Python packages) to implement all the mentioned models and algorithms, the simulators/functions used to generate the studied data are of the public domain and referenced accordingly. Alongside a thorough and clear paragraph on the experimental settings, we also attach a compressed file containing the source code used to reproduce the presented experiments. In the end, we commit to provide a non-anonymized dynamic link to the repository with the most up-to-date version of the code soon.

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

# A APPENDIX

Table 3: Benchmark of acquisition functions on test sets. The table shows the mean and standard deviation over 10 runs.

| Simulator | Iter. | | GP (MAP) ALM | FBGP (MCMC) ALM | B-ALM | B-QBC | QB-MGP | BALD |
|---|---|---|---|---|---|---|---|---|
| Sine | 30 | NLML | 1.34 ± 0.39 | 0.28 ± 0.08 | 0.29 ± 0.11 | **0.21 ± 0.09** | 0.26 ± 0.14 | 0.40 ± 0.11 |
| | | RMSE ($\mu$) | 0.69 ± 0.17 | **0.15 ± 0.03** | **0.15 ± 0.02** | 0.23 ± 0.08 | 0.16 ± 0.02 | 0.34 ± 0.22 |
| Gramacy1d | 80 | NLML | -0.25 ± 0.03 | -0.23 ± 0.03 | -0.24 ± 0.03 | -1.12 ± 0.11 | **-1.18 ± 0.08** | -0.09 ± 0.42 |
| | | RMSE ($\mu$) | 0.33 ± 0.01 | 0.32 ± 0.00 | 0.32 ± 0.00 | 0.08 ± 0.02 | **0.07 ± 0.01** | 0.37 ± 0.19 |
| Higdon1d | 50 | NLML | -0.12 ± 0.17 | -0.02 ± 0.09 | -0.01 ± 0.13 | **-0.31 ± 0.05** | -0.19 ± 0.17 | 0.13 ± 0.18 |
| | | RMSE ($\mu$) | 0.09 ± 0.03 | 0.10 ± 0.02 | 0.10 ± 0.02 | **0.08 ± 0.01** | 0.08 ± 0.03 | 0.30 ± 0.19 |
| Motorcycle | 20 | NLML | 0.92 ± 0.52 | 0.71 ± 0.13 | 0.65 ± 0.07 | 1.61 ± 0.79 | **0.63 ± 0.08** | 1.04 ± 0.76 |
| | | RMSE ($\mu$) | 0.26 ± 0.06 | 0.28 ± 0.07 | 0.28 ± 0.07 | 0.90 ± 0.26 | **0.24 ± 0.05** | 0.38 ± 0.22 |
| Gramacy2d | 30 | NLML | 2.27 ± 0.77 | 1.53 ± 0.61 | 1.03 ± 0.18 | 1.32 ± 0.73 | **0.69 ± 0.33** | 1.79 ± 0.51 |
| | | RMSE ($\mu$) | **0.09 ± 0.01** | 0.10 ± 0.01 | 0.10 ± 0.01 | 0.10 ± 0.02 | 0.11 ± 0.01 | **0.09 ± 0.01** |
| Branin2d | 30 | NLML | -0.33 ± 0.16 | -0.24 ± 0.19 | -0.14 ± 0.15 | **-0.39 ± 0.11** | -0.22 ± 0.13 | 0.36 ± 0.31 |
| | | RMSE ($\mu$) | 42.16 ± 2.14 | 42.29 ± 1.37 | 42.25 ± 1.83 | 42.61 ± 1.32 | **41.48 ± 1.26** | 46.58 ± 6.94 |
| Ishigami3d | 100 | NLML | 0.98 ± 0.26 | 0.94 ± 0.27 | 0.60 ± 0.46 | -0.27 ± 0.24 | 0.36 ± 0.52 | -0.50 ± 0.16 |
| | | RMSE ($\mu$) | 7.24 ± 0.98 | 6.05 ± 0.80 | 5.50 ± 0.45 | 5.29 ± 0.17 | 5.34 ± 0.22 | **4.50 ± 0.34** |
| Hartmann6d | 100 | NLML | -1.80 ± 0.08 | -1.44 ± 0.18 | -1.12 ± 0.39 | **-2.12 ± 0.18** | -1.63 ± 0.36 | -1.29 ± 0.28 |
| | | RMSE ($\mu$) | 1.18 ± 0.01 | 1.18 ± 0.01 | 1.17 ± 0.01 | 1.21 ± 0.01 | 1.19 ± 0.01 | **1.14 ± 0.04** |
| **Best** | | | 1/16 | 1/16 | 1/16 | 5/16 | **7/16** | 3/16 |

