# OpenReview forum: "Bayesian Active Learning with Fully Bayesian Gaussian Processes"
_ICLR.cc/2022/Conference — ICLR 2022 Submitted_

### Official Review · Reviewer_uC7q · 2021-10-29

**Correctness:** 3
**Technical Novelty And Significance:** 3
**Empirical Novelty And Significance:** 2
**Recommendation:** 5
**Confidence:** 4

**Main Review:**

**Strengths**

- the contributions of this paper are interesting and appear to be original
- the approaches are based on sound probabilistic foundations
- the experiments on synthetic data provide intuitive insights into the relative merits of different approaches.

**Weaknesses**

- connections and comparison to prior work is incomplete
- experimental results are not very conclusive; no experiments on real data
- parts of the paper are unnecessarily convoluted and unclear

The problem of overfitting to the marginal likelihood problem is well-known in the GP community. In the context of Bayesian optimization for example, some authors have proposed a similar method that takes advantage of a mixture of GPs [1]. How does this paper's approach differ?

The experimental results on synthetic data are quite comprehensive and insightful. One relevant baseline that appears to be missing is Zhao et al (2020). How does their approach compare empirically?

It is difficult to reconcile the results on the negative log-marginal likelihood (NLML) with the RMSE. The squared error is identical to the negative log-likelihood (under a Gaussian model) and should thus closely match the NLML results - how do you explain the difference in relative ranking and the fact that many of the RMSE curves slope upwards? How is the RMSE in fact evaluated—is there a hold-out set?

Additional experiments on real-world datasets & finding one realistic application where the proposed approach is clearly superior would significantly strengthen the paper.

Finally, I encourage the authors to improve the clarity of the paper: introducing all concepts progressively, and making statements precise. Examples:

- canonical Gaussian Process: to the best of my knowledge, this is not a widely used terminology for the RBF kernel
- metamodeling is never formally introduced
- "given the data (x, y), ..., a GP is typically denoted as GP(...)" -> the data does not appear to be relevant to the way the GP is denoted.
- the definition of the RBF kernel is incorrect (there is a scalar quantity, the squared norm, that is divided by a vector)
- I don't think the reference to the "no free lunch" theorem is appropriate in the context of the discussion on page 4
- "the inner integral is intractable and is therefore optimized with approximate inference": what exactly is optimized?
- section 5: how is convergence declared?

[1]: An Empirical Bayes Approach to Optimizing Machine Learning Algorithms, J. McInerney, NIPS 2017

**Summary Of The Paper:**

This paper introduces two new active-learning strategies for Gaussian process regression, based on a fully-Bayesian treatment of GPs. Instead of using fixed hyperparamers obtained by maximizing the marginal log-likelihood of the GP model, the authors suggest learning a posterior belief over GP hyperparameters, and make use of the full distribution to design more effective active-learning methods.

In practice, this results in two concrete proposals:

1) a strategy that is similar to query-by-committee, where the committee is formed of multiple GPs sampled from the hyper-posterior (B-QBC).
2) a variant that combines a notion of disagreement with the predictive uncertainty (QB-MGP)

These methods are then compared to others on 8 synthetic datasets.

**Summary Of The Review:**

I think there is potential in this paper: the problem is important and the method is sound. Improving the clarity of the paper, comparing against McInerney (2017) and Zhao et al. (2020) and applying the method on real-world data would make this paper significantly stronger.

---

> ### Author Response · Authors · 2021-11-23
> **Response to review**
>
> Thank you for your time and comments.
>
> Thank you for referring to relevant literature. We will consider these papers in the next version of the paper.
>
> It is true that the negative log likelihood (NLL) is similar to RMSE but not the negative log marginal likelihood (NLML). The likelihood measures the probability of the target under the model. In contrast, the marginal likelihood measures how likely the targets are from all the underlying noise-free models given the data. They are therefore not expected to give the same results. Some of the RMSE curves go upwards because the data is better described by a mean rather than a noisy mean-function. We will elaborate on this in the next version of the paper. Thank you for pointing out that we are missing a description of the test set. The acquisition functions are all evaluated on a separate test set with 2000 data points.
>
> We agree and will consider the opportunities of including a real-world problem.
>
> Thank you for pointing out examples that need to be explained better. We will consider all these cases when we make the next version of the paper.

---

> > ### Comment · Reviewer_uC7q · 2021-11-30
> > **Acknowledgment of response**
> >
> > Thank you for your response, and in particular for your clarifications about the NLML and RMSE—indeed I was mistaken and they are not identical.

---

### Official Review · Reviewer_ssFN · 2021-11-02

**Correctness:** 4
**Technical Novelty And Significance:** 3
**Empirical Novelty And Significance:** 2
**Recommendation:** 6
**Confidence:** 4

**Main Review:**

The paper seems to be interesting.
The part that I most enjoy to read is Section 2 of "related works" and Section 4.1, which are also the clearer parts of the paper.

- The rest of the paper, including abstract and introduction, is quite confusing in my opinion. The concepts of "bias-variance trade-off" and "active learning" are often referred but it is very difficult to understand the connection.

- Some sentences like:

"In this paper, we follow a general approach and assume no prior knowledge about the kernel or
hyperparameters. We consider multiple model hypotheses by replacing the fitting procedure of the
marginal likelihood with Markov Chain Monte Carlo (MCMC) sampling. Lalchand & Rasmussen
(2020) show that with fixed, medium-sized data sets and carefully chosen kernels, it is beneficial
to fit GPs with MCMC instead of by maximizing the marginal likelihood."

are not very precise. I assume that you compare different kernels approximating the marginal likelihood ?
Or maybe you refers considering a mixture of models with different kernels. In any case, we are assuming some kind of kernel functions and you have to learn their parameters. Please, improve your explanations in all the work.

- Regarding the state of the art discussion in Sections 2 and 4.1 about active learning. Novel more robust (with less numerical problems) and more advanced (considering the gradient of the solutions) acquisition functions has been considered in the literature

D. H. Svendsen et al, Active Emulation of Computer Codes with Gaussian Processes - Application to Remote Sensing, Pattern Recognition Volume 100, 2020,

F. Llorente et al., "Adaptive quadrature schemes for Bayesian inference via active learning", IEEE Access, Volume 8, 2020,

M. Kanagawa and P. Hennig, “Convergence Guarantees for Adaptive Bayesian Quadrature Methods,” in Advances in Neural Information Processing Systems, 2019, pp. 6234–6245.

They have been successfully applied in regression and to build adaptive quadrature schemes (all of them in the context of active learning). Please, include these kind of acquisition functions in your discussion.



**Summary Of The Paper:**

It seems that the authors introduces two novel acquisitions functions for Gaussian Process models:
the first one is a mode-seeking Bayesian variant of Query-by- Committee (B-QBC), and the second is simultaneously mode-seeking and minimizing the predictive variance through a Query by Mixture Gaussian Processes (QB-MGP) formulation.  The authors also present several simulations.

**Summary Of The Review:**

The paper contains interesting material, but specially abstract and introduction are very confused.
The state-of-the-art discussion must be also improved.

---

> ### Author Response · Authors · 2021-11-23
> **Response to review**
>
> Thank you for your time and comments.
>
> We will make the connection between active learning and the bias-variance trade-off more explicit in the next version of the paper. We will also make our statements more precise. Regarding the particular sentence, we would like to also elaborate here:
>
> We assume no prior knowledge about the functional form of the data. If one knows that the data has a periodic or a linear trend, one should use a periodic or linear kernel, respectively. Here, we have no prior knowledge about the data, so we use the general-purpose kernel, RBF. Likewise, we know nothing about the hyperparameters of the kernel (in this case, the length scale and the noise), and thus we apply vague priors to these hyperparameters. This is in contrast to the case where you do have information about the data, e.g., if you happen to know that the data have high variance, you should apply a strong prior for high noise. Often, you will fit these hyperparameters by a point estimate (e.g., MAP through the marginal likelihood), but here we use MCMC to get their joint posterior. The result is that we have multiple models (same kernel, but different hyperparameters), which represents different model hypotheses.
>
> Thank you for referring to relevant literature. We will consider these papers in the next version of the paper.

---

### Official Review · Reviewer_H5XZ · 2021-11-02

**Correctness:** 3
**Technical Novelty And Significance:** 3
**Empirical Novelty And Significance:** 2
**Recommendation:** 5
**Confidence:** 4

**Main Review:**

Strengths: This work is interesting and well-motivated. Bayesian Gaussian process-based active learning has practical benefits as demonstrated in numerical tests.

Weaknesses:
1. It is well known that active learning tends to explore the function space. The proposed B-QBC seems to exploit more of the available information. Is there any intuition why it works well in practice?
2. It is not clear what the performance metrics (i.e., NLML and NMSE) are in the experiments. How is the experimental validation different from Bayesian optimization?
3. The experiments are based on synthetic functions. It would be better if practical problems are considered.

**Summary Of The Paper:**

This paper considers Bayesian treatment of the hyperparameters in Gaussian process based active learning, based on whichtwo novel acquisition functions are proposed.

**Summary Of The Review:**

This is an interesting work that considered Bayesian Gaussian processes for active learning. But the merits of the paper can be better enhanced by addressing the aforementioned comments.

---

> ### Author Response · Authors · 2021-11-23
> **Response to review**
>
> Thank you for your time and comments.
>
> In the following, we address the three comments.
>
> 1. The intuition is that B-QBC queries data points that guide the model towards the correct hyperparameters. This might sound trivial, but the often used acquisition function based on the entropy focuses more on exploring the input space (here, through the predictive uncertainty) rather than finding the optimal hyperparameters first.
>
> 2. The negative log marginal likelihood (NLML) is calculated as $p(y|X) = \int p(y|f,X)p(f|X)dx$ and the root mean square (RMSE) as $RMSE = \sqrt{\dfrac{1}{N}\sum_{i=1}^N(y_{pred}-y_{target})^2}$. The metrics are evaluated on a separate test set.
>
> 3. In Bayesian Optimization, the goal is to find an optimal value; whereas in active learning, the goal is to learn a predictive model with the fewest possible data points.
>
> 4. We agree and will consider the opportunities of including a real-world problem.

---

### Official Review · Reviewer_hDWW · 2021-11-02

**Correctness:** 2
**Technical Novelty And Significance:** 3
**Empirical Novelty And Significance:** 2
**Recommendation:** 3
**Confidence:** 3

**Main Review:**

####################################

Strengths:
- The themes discussed in this paper, including the full posterior having modes corresponding to either signal or noise which should be discriminated between with active learning, are very interesting. In particular, Figures 1 and 2 were interesting and illustrative.
- The related work section and description of Gaussian Processes is mostly satisfactory
- The experiments compare against a diverse set of active learning acquisition functions and experimental simulators

####################################

Weaknesses:

While this paper presents some interesting ideas, overall I feel that certain major elements need to be significantly improved before it is ready for publication.

When I first read the abstract, I was intrigued by the idea of the bias-variance tradeoff being neglected in active learning, how in GP this tradeoff corresponds to the length scale and noise term, how “the optimal mode of the joint posterior of the hyperparameters is equivalent to the optimal bias-variance tradeoff”, and how B-QBC and QB-MGP directly incorporate this tradeoff in their acquisition functions to mitigate bias-variance issues. However, in my opinion, these points are not strongly reflected in the paper itself:

*** Introduction ***

Starting in the introduction, I had expected a cohesive theme illustrating the issues presented in the abstract: how the bias-variance tradeoff is neglected in active learning, a strong argument for bias-variance correspondence to length scale and noise term, and strong connections with the hyperparameters posterior. Instead, I found the topics in the introduction only very loosely coupled: specifically, I think the connections between the predictor vs guide, bias vs variance, and signal vs noise in GPs need to be strengthened and more direct arguments made. For instance, statements such as “The problem is directly related to the well-known bias-variance trade-off, although we reformulate it as a balance between modeling the data as signal or noise” or “However, none of
these approaches directly address the core problem: which mode to choose? Ideally, this should be
answered with prior information about the problem, although typically that is not available, making
these approaches less practical.” seem extremely important, but most of the time I found such critical statements vague and unsupported. While Figure 1-2 were interesting, I did not find myself convinced that the bimodal behavior in Figure 2 is a general phenomenon and found myself unsure how it connected to bias and variance. Is this multimodal behavior between signal and noise an established fact in the literature, or is this perspective a major contribution of this paper? Either way, I believe statements around this point need to be strengthened. Overall, after reading the introduction I found myself unconvinced about the strong points presented in the abstract. I also expected that I would find more details supporting these points in Section 3, which seem to be missing there (e.g., establishing connections between bias-variance and specific GP hyperparameters).

*** Sections 2-3 ***

Otherwise, I was mostly happy with Sections 2-3. One point in Section 3 I’d like to bring up is that the authors claim earlier in the paper that “In this paper, we follow a general approach and assume no prior knowledge about the kernel or hyperparameters.” However, the authors state in Section 3 that they use an RBF kernel. Can the authors comment about this? Is this specific choice not in fact an assumption about the kernel?

*** Section 4 ***

In Section 4, B-QBC seems reasonable, but I was not convinced that it somehow addressed the key bias-variance tradeoff and/or mode selection problem in GP active learning described earlier in the paper. The idea of discriminating between modes seems reasonable, but I believe that in order to motivate B-QBC, a much stronger argument (stronger than the illustrative example in Figures 1-2) needs to be presented about why the hyperparameter posterior is multimodal, and how B-QBC will specifically select the correct one. In terms of QB-MGP, I found myself confused by its presentation. I was not convinced that equation (11) is maximizing the predictive variance of the mixture model --- I can imagine that it comes directly from a variance decomposition, but I believe the derivation of this fact should be presented. Moreover, as with B-QBC, why should maximizing the predictive variance of this mixture model solve the mode-selection problem and tradeoff between bias and variance? Furthermore, how do the authors expect B-QBC and QB-MGP to compare qualitatively? Since QB-MGP is equivalent to B-QBC plus a variance term, how do the authors expect this variance term to impact the performance of the acquisition function? Overall, B-QBC and QB-MGP seem to me to be the two fundamental contributions of the paper, and I believe more discussion is needed here to derive and motivate their use and strongly connect them to resolving bias-variance/signal-noise/mode selection tradeoffs.

*** Section 5 ***

While I appreciate the multitude of simulators tested in the experiments, I am not convinced by the experimental results. The performance curves seem to have high variance, and I am not convinced by the summary statistics and takeaways presented. It is not at all clear to me from Figure 4-5 and Tables 2-3 that “B-QBC outperforms the benchmark functions, whereas QB-MGP is the most robust acquisition function and achieves the best accuracy with the fewest iterations.”

Specifically, analyzing performance at a single iteration number based on a converged reference curve only captures a small part of the overall picture. I believe it would be better to examine summary statistics that better capture the overall performance of each method, such as evaluating multiple iteration numbers or looking at summary statistics such as AUC. The authors make a comment in section 5.3 attempting to justify this single iteration point evaluation, but I did not find it convincing. I am also unsure if the likelihood ratio and error quotients in Table 2 provides an adequate summary of the data. When using the raw results in Table 3, any sort of benefits for B-QBC and QB-MGP in Figure 5 become much less clear to me when looking at raw data instead of quotients. Can the authors comment about this choice of presenting summary statistics in terms of relative performance rather than raw values? Regardless, given the not insignificant error bars, I would be more convinced by these results if they were accompanied by statistical testing against either the best performing method or random sampling (see other active learning meta-analyses such as “A benchmark and comparison of active learning for logistic regression”, Yang and Loog 2018).

In Figure 4, it seems difficult to take away anything very useful from these experiments, due to the overlapping error bars in each trial and the fact that in some experiments and methods (including QB-MGP and B-QBC) the error is in fact increasing. It seems that the performance differences can be made more clear by plotting error bars for summary statistics (e.g., standard error or confidence intervals) and increasing the number of trials beyond only 10. I do not look at Figure 4 and have confidence that I should use QB-MGP and B-QBC over the other methods. In my opinion, the takeaways stated in the text are overstated based on these curves, such as QB-MGP being the “best choice” from the Sine RMSE plot, when it seems to be that multiple methods tie QB-MGP in performance, or that B-QBC could even be considered better than the other methods, given the overlapping error bars. Furthermore, there are analytical takeaways that to me seem vague and unsupported, such as “This agrees with our expectation that QB-MGP seems to be more robust than B-QBC to different types of noise.” Why is this an expectation? Statements such as “QB-MGP is achieving the highest RMSE due to favoring to capture the small region with a non-linear signal compared to the vast linear region” or “Looking at all the active learning iterations, the GP fitted with MAP seems to give the best trade-off between the NLML and RMSE, indicating that the regular GP with ALM is the best for smooth problems” are in my opinion speculative takeaways that seem unsupported. Could QB-MGP favoring the non-linear portions be shown empirically, for instance?

Although it’s possible that I missed these details in my reading (although I went back again to check), I have some reproducibility concerns. It is unclear how “convergence” is calculated for selecting each plot reference as well as the examined iteration number. How exactly are NLML and RMSE calculated, do they utilize the full posterior, and which data are they calculated with respect to --- the sampled data points, or a held-out set of randomly sampled data? Are the distributions in Figure 5 at the simulator level, or is all data pooled together from all simulators? In Table 2, if different trials' references converged at different times, how was a single iteration value selected? Low-level details such as these are important and seem to be missing (or should at least be included in an appendix), which prevents a full understanding of the experimental results.

Furthermore, I believe additional experiments should be run to support the main claims made in the paper. A major claim is that the full posterior is multi-modal, motivating the proposed active learning experiments. I think an experiment that shows this empirically through some metric would strengthen this major claim. I also think that additional experiments are needed to characterize the performance of the proposed schemes beyond performance alone. Specifically, my reading of the paper is that the proposed methods are designed to select the correct full posterior mode as quickly as possible. It would be useful to show during trials of B-QBC and/or QB-MGP how a single (correct) mode of the full posterior emerges during MCMC sampling, and the others are suppressed (and possibly comparing this metric against baseline methods).

In general, I did not find myself convinced by Section 5.3 and found it lacking. It discusses two important details --- why a single iteration slice is sufficient for performance analysis, and where specifically in the acquisition and model evaluation pipeline the full posterior was used rather than the hyperparameter MAP (and why this choice was made for every step) --- but I found the first discussion unconvincing and the second discussion a bit  confusing. Specifically, I found this important sentence confusing “However, since we focus on benchmarking the acquisition functions and not the models, we believe that using the mode is more accurate when comparing it to the regular GP since we only change the fitting procedure.”

######################################

Other points *not* factoring into decision, but serving as feedback for the authors:

It has been pointed out before (Settles, B. (2012). Active learning: Synthesis lectures on artificial intelligence and machine learning. Long Island, NY: Morgan &  Clay Pool. --- Section 3.5) that information gain maximization can be interpreted as a type of query by committee, where KL divergence on the label is used as a notion of disagreement. Specifically, if the BALD acquisition function is equivalently written as $\mathbb{E}_{\theta} [\mathrm{KL}(p(Y \mid \theta) \Vert p(Y))]$, then this is comparable to B-QBC, which uses a mean discrepancy instead of a KL-divergence. I think this is an interesting connection that would strengthen the work in terms of why B-QBC might perform better than BALD. What is mean discrepancy doing well that KL divergence isn't?

To me, the use of "GMM" seems unusual here to describe a mixture of Gaussian Processes. I checked the literature cited by the authors, and I did not see “Gaussian Mixture Model” or GMM being used as the authors do here to describe a mixture of Gaussian Processes (as opposed to the classical GMM notion of a mixture of multivariate Gaussians in a vector space). Personally, I think using the phrase "GMM" runs the risk of confusing the model with a classical GMM framework (as opposed to a Gaussian Process). However, this may be a matter of personal opinion.

The first paragraph of Section 5 seems out of place to me. It discusses important points, but I think Section 5 should start with a more general description of the experiments instead of just the Sine simulator. Also, I think the experiments could be strengthened by comparison against random (passive) sampling.

Minor points:
- in the abstract, Query by Mixture Gaussian Processes -> Query by Mixture *of* Gaussian Processes
- the sentence “The literature suggests handling this problem with clever initializations...either always initializing…” seems redundant to me
- the location of Gramacy2d and Motorcycle should probably be switched in Figure 4 since the text seems to follow a row-first description of Figure 4
- I think the right subplot of Figure 5 should show a zoomed-in region to better visualize the methods besides BALD
- Personally, I don’t think the future work paragraph adds much to the paper, and I don’t understand what this sentence proposes: “Eventually, our ultimate goal is to provide practitioners with an auxiliary tool to map the simulators’ output behaviors in a more efficient manner.”


**Summary Of The Paper:**

This paper takes a fully Bayesian approach to Gaussian Process (GP) active learning by using MCMC sampling to consider multiple model hypotheses from a full posterior, from which it selects examples for GP regression using two new active learning strategies --- Bayesian Query-by-Committee (B-QBC) and Query by Mixture of Gaussian Processes (QB-MGP). B-QBC queries the data point that maximizes the disagreement between the sampled models’ mean values. The idea is that this should provide the highest information about the optimal posterior mode. QB-MGP uses a combination of B-QBC with entropy sampling. The authors evaluate these strategies against several baseline active learning methods on several simulators.

**Summary Of The Review:**

In my opinion, this paper is ultimately making strides at an interesting and important topic in GP active learning --- what role does the full posterior have in managing the bias-variance tradeoff, and how can these insights be used to design better active learning acquisition functions? I think the proposition of QB-MGP and B-QBC is an important stride in this direction. However, I believe the paper is not ready for publication. In terms of presentation, I found the arguments made about bias-variance in its relation to the full posterior and the motivation for QB-MGP and B-QBC unconvincing. I also found some statements made in the paper to be confusing or unsupported. In terms of experiments, I found myself unconvinced by the presented analysis (specifically due to the high variance in the results and the lack of more general summary statistics such as AUC rather than a single iteration slice) and am concerned with some aspects of reproducibility. Therefore, I would recommend **rejection** for this paper and encourage the authors to revise the writing to strengthen the presented arguments, better motivate their active learning strategies, improve reproducibility, and conduct a stronger analysis and visualization of the experimental results.

---

> ### Author Response · Authors · 2021-11-23
> **Response to review**
>
> Thank you very much for your time and comprehensive review. We are happy to see that you like the overall idea of utilizing the joint posterior of the hyperparameters in the active learning procedure. We can definitely recognize that we could have been more clear about the connection between the hyperparameters, the bias-variance trade-off, and the posterior. In the next version of this work, we will strengthen the argumentation and make these connections explicit.
>
> We will also use a metric to both empirically prove and track the multi-modal behavior (e.g. entropy and kernel density estimations to find modes). We believe that this will make the motivation stronger and clearly show the behavior/benefits of B-QBC and QB-MGP.
>
> In the experiment section, we had missed that we had not included a description of the test set. We use a separate test set of 1000 data points, and we will include a description next time. We will define a precise measure of convergence, too. In the literature, stopping criteria in active learning is a research field on it is own, however, we understand that a more concise way of measuring convergence is needed. We will also evaluate the acquisition functions by either considering multiple iterations or a measure like AUC. Thank you for the references.
>
> Thank you for the final points on QBC, GMM, and the structure, as well as the minor comments. We will take all comments into account in the next version.
>
> We appreciate your time and your in-depth review, it is quite impressive. If you happen to swing by Copenhagen, we will buy you a beer :-)
>
> PS. A small comment regarding the literature:
> Finding literature that says that "the bias-variance trade-off is neglected in active learning" and "prior information is typically not available" is not straightforward. On the other hand, it is not easy to find any paper considering the bias-variance trade-off in the process of active learning. Likewise, only a small subset of the papers in the literature have prior knowledge available. Regarding the existence of multi-modal posteriors for a GP, this paper [1] investigates that problem.
>
> [1] Yao, Yuling, Aki Vehtari, and Andrew Gelman. "Stacking for non-mixing Bayesian computations: The curse and blessing of multimodal posteriors." arXiv preprint arXiv:2006.12335 (2020).

---

> > ### Comment · Reviewer_hDWW · 2021-11-29
> > **Thanks for your response**
> >
> > Thank you for your response and the pointers to the literature!

---

### Decision · Program_Chairs · 2022-01-20

**Decision:**

Reject

**Comment:**

This paper studied Bayesian active regression with Gaussian processes, and proposed two intuitive algorithms inspired by the classical disagreement-based and uncertainty sampling criteria. The reviewers appreciate the motivation and overall idea of taking a fully Bayesian approach by utilizing the joint posterior of the hyperparameters for active learning. However, there are shared concerns among the reviewers in the clarify and consistency of several key technical components, including discussion around bias-variance tradeoff and its connection to the fully Bayesian approach, as well as in the experimental details, which make the current package insufficient for publication.

Reviewers provide very useful feedback (in particular with a very extensive review by Reviewer hDWW) for improving the current work. The authors acknowledge in their responses that these are valid concerns and they would address these issues in a further version of this work.